# The Utility of High Intensity Interval Training to Improve Cognitive Aging in Heart Disease Patients

**DOI:** 10.3390/ijerph192416926

**Published:** 2022-12-16

**Authors:** Jenna L. Taylor, Jill N. Barnes, Bruce D. Johnson

**Affiliations:** 1Human Integrative and Environmental Physiology Laboratory, Mayo Clinic, Rochester, MN 55902, USA; 2Department of Cardiovascular Medicine, Mayo Clinic, Rochester, MN 55902, USA; 3Bruno Balke Biodynamics Laboratory, Department of Kinesiology, University of Wisconsin-Madison, Madison, WI 53706, USA

**Keywords:** exercise, cardiac rehabilitation, cardiovascular disease, heart failure, brain, cerebral blood flow, cerebrovascular function, cognitive function, gray matter, white matter

## Abstract

Adults with cardiovascular disease and heart failure are at higher risk of cognitive decline. Cerebral hypoperfusion appears to be a significant contributor, which can result from vascular dysfunction and impairment of cerebral blood flow regulation. In contrast, higher cardiorespiratory fitness shows protection against brain atrophy, reductions in cerebral blood flow, and cognitive decline. Given that high intensity interval training (HIIT) has been shown to be a potent stimulus for improving cardiorespiratory fitness and peripheral vascular function, its utility for improving cognitive aging is an important area of research. This article will review the physiology related to cerebral blood flow regulation and cognitive decline in adults with cardiovascular disease and heart failure, and how HIIT may provide a more optimal stimulus for improving cognitive aging in this population.

## 1. Introduction

Dementia is a debilitating disease that impairs a person’s memory and cognitive functions for maintaining independence and normal daily activities. Mild cognitive impairment is a term used to characterize individuals that fall between the cognitive changes of normal aging and dementia but still have the ability to perform usual daily activities and live in an autonomous manner [1]. Globally, it is estimated that around 47 million people are living with dementia-related disease, which is expected to triple by the year 2050 [2]. Vascular dementia and the neurodegenerative condition Alzheimer’s disease are the two most common causes of dementia. Although Alzheimer’s disease has distinct pathology from vascular dementia (e.g., β-amyloid plaques and neurofibrillary tangles of tau protein), they frequently coexist and can simultaneously contribute to cognitive impairment [3]. Moreover, 30% of Alzheimer’s disease cases have been attributed to modifiable vascular risk factors such as hypertension, obesity, diabetes, physical inactivity [4].

Cardiovascular disease (CVD) affects >400 million people worldwide [5] and 120 million people in the United States alone [6]. While the incidence of dementia increases with age, adults with CVD have a higher risk of cognitive impairment [7,8]. Furthermore, a cardiovascular event or the onset of heart failure (HF) accelerates the cognitive decline [9,10]. Reduced blood flow to the brain (i.e., cerebral hypoperfusion) appears to be a significant contributor to cognitive decline [11]. Although reductions in cerebral blood flow (CBF) occur with normal aging [12,13] the age-related decline may be exacerbated by the presence of CVD. Indeed, patients with coronary artery disease have shown reduced CBF and greater rates of brain atrophy compared with age-matched controls [14,15]. Due to medical advancements, patients with CVD are now living for many decades [16]. Therefore, the impact of CVD on the aging brain and cognitive decline is of increasing importance.

This article will provide an overview of the pathophysiology that contributes to cerebral hypoperfusion and cognitive decline in adults with CVD including HF, and potential mechanisms for how exercise promotes better brain health. Furthermore, this article will review the current evidence on whether exercise training may improve brain health in patients with CVD, and how high intensity interval training (HIIT) may provide a more optimal stimulus.

## 2. Cerebral Blood Flow Regulation

Adequate CBF as well as the structural and functional integrity of cerebral blood vessels are essential to normal brain functioning [17]. Reduced CBF and impaired cerebrovascular function represent early physiological markers of neurocognitive disorders that proceed physical signs and symptoms [17]. The term cerebrovascular function has been used broadly to encompass resting levels of CBF, cerebrovascular conductance (CBF/mean arterial blood pressure (MAP)), cerebrovascular resistance (MAP/CBF), and indices of CBF regulation [18]. The regulation of CBF is characterized within the literature in several ways. Cerebrovascular reactivity characterizes the CBF response to changes in carbon dioxide (CO_2_), whereby cerebral vessels should vasodilate and increase CBF with elevated levels of arterial CO_2_ (hypercapnia); and vasoconstrict and decrease CBF with reduced levels of arterial CO_2_ (hypocapnia) [13,19]. Neurovascular coupling involves CBF response to changes in neural activity (e.g., visual stimuli, cognitive tasks) [20,21]. Within each neurovascular unit, modulations in neuronal activity alter the smooth muscle vascular tone and local blood flow either directly or mediated through the astrocyte glial cell within the unit [21]. Cerebral autoregulation is defined as the intrinsic ability to maintain CBF in response to changes in perfusion pressure, predominately influenced by arterial blood pressure [19]. While there has been extensive debate regarding the understanding of cerebral autoregulation in humans, contemporary views promote further characterization as static autoregulation based on CBF response under steady-state conditions (e.g., >10 min); or dynamic autoregulation based on CBF response to spontaneous (resting) or induced (e.g., Valsalva, sit-to-stand, exercise) fluctuations in blood pressure [22,23]. Finally, there appears to be a systemic regulation of CBF with changes in cardiac output, independent of changes in arterial blood pressure [24,25,26]. Non-neural flow-mediated regulatory mechanisms have been suggested, whereby changes in pulsatile flow and shear stress alter nitric oxide release or endothelin production, which in turn influences cerebrovascular resistance and CBF [27].

## 3. Cardiovascular Disease and Brain Health

Several studies have shown that the risk of cognitive impairment is increased for patients with CVD or HF [7,8,28,29]. Meta-analyses have estimated the risk of cognitive impairment is 62% higher in HF and 45% higher in coronary heart disease [7]. The Women’s Health Initiative Study found in postmenopausal women, that those with a previous myocardial infarction had the greatest risk of cognitive impairment (HR: 2.10, 95%CI: 1.40–3.15) compared with any CVD (HR: 1.29, 95%CI: 1.00–1.67) [29]. Furthermore, the Swedish Twin Study showed that increased risk of cognitive decline with CVD is not isolated to vascular-related dementias, with CVD doubling the risk of Alzheimer’s disease in adults with genotype predisposition (*APOE4* allele carriers) [30].

Dementia and CVD share common risk factors (such as hypertension, adiposity, hyperlipidemia, and diabetes mellitus), that have adverse effects on vascular structure and function [31]. Hypertension and obesity are both associated with hyperactivity of the sympathetic nervous system [32,33], which can result in excessive vascular resistance, arterial stiffness, and adverse cerebrovascular remodelling [34]. Excess adiposity, hyperlipidemia, and hyperglycemia also adversely affect the vasculature and brain through inflammatory pathways [35,36]. Inflammation and oxidative stress damages the endothelial cell layer [31], which through release of nitric oxide, plays an important role in vascular function and maintaining the integrity of the brain blood barrier [37,38,39]. Alterations in vascular structure and tone can modify cerebral hemodynamics and lead to chronic reductions in CBF, disrupted regulation of CBF, and impaired protein clearance [40,41]. Adequate CBF is important for maintaining the integrity of the blood brain barrier. Cerebral hypoperfusion, and subsequently reduced oxygen and glucose to the brain, can negatively affect the endothelial and neighboring cells within the neurovascular unit [40,42]. Reduced glucose to the brain can induce excitotoxicity and cause metabolism disruption, mitochondrial dysfunction, activation of proteases/phospholipases, and production of reactive oxidative species, which collectively disrupt blood brain barrier integrity through cell membrane damage and vascular cell death [42]. Breakdown of the blood brain barrier can make neurons susceptible to damage by facilitating the entry of neurotoxic compounds and pathogens that initiate neurodegenerative pathways [43]. Reductions in cardiac function, which decreases cardiac output and systemic perfusion, may also contribute to cerebral hypoperfusion and cognitive impairment [44]. The relationship between cardiac output and middle cerebral artery blood velocity (MCAv) appears to be linear [26,45]. The extent of CBF reduction in patients with HF has been shown to correlate with disease severity [46], and increases in MCAv have been found with improvement in left ventricular ejection fraction following cardiac resynchronization therapy [47]. Although, Hammond et al. [10] found that cognitive decline was similar in HF patients with reduced versus preserved ejection fraction. In HF patients with preserved ejection fraction, the mechanisms for cognitive decline are likely related to obesity, vascular impairment, diastolic dysfunction, chronic neurohormonal activation [48].

Large artery stiffening is another link between CVD and the brain. Arterial stiffness and wave reflection (the reflection of the pressure pulse wave back to the heart), increases pulse pressure and pulsatile blood flow to the smaller vessels of the arterial tree (notably the brain) [49]. This increased pulse pressure contributes to microvascular pulsatility and hemodynamic stress within the perivascular spaces of the brain, that results in microstructural damage and impairments in β-amyloid clearance [50]. Arterial stiffness is a common cause of cerebral small vessel disease, which affect the arteries, arterioles, capillaries, and veins within the brain [51]. Two common types of cerebral small vessel disease include (1) arteriosclerosis (referred to as arterial stiffening or hypertensive small vessel disease), characterized by the loss of smooth muscle cells, narrowing of the lumen, and thickening of the vessel walls; and (2) cerebral amyloid angiopathy, characterized by the progressive accumulation of β-amyloid protein [51]. This damage to the cerebral vessels can cause ischemia, inflammation, vessel rupture, and disruption of the brain-blood barrier and neural connectivity pathways, that results in pathological brain changes such as white matter hyperintensities, lacunar infarcts, microbleeds, and macroscopic hemorrhage [51,52].

Coronary artery bypass graft surgery (CABG) has been proposed to increase risk of cognitive decline due to intraoperative hypotension, hypoxia, microembolism, and/or inflammatory processes [53]. However, a study by Sweet et al. [54] compared patients following CABG or percutaneous coronary intervention (PCI) with healthy controls, and found a similar degree of decline on neurocognitive tests for both CABG and PCI over 3-weeks, 4-months, and 12-months. Furthermore, Selnes and colleagues conducted an expert review of the topic and concluded ‘the extent of pre-existing cerebrovascular and systemic vascular disease’ have a greater effect than procedural variables on neurocognitive function in the short- and long-term [55].

## 4. Mechanisms for Improving Brain Health with Exercise

There is substantial evidence that exercise training is an effective way to increase cardiorespiratory fitness [56], improve vascular structure and function [38,57], and reduce traditional CVD risk factors and inflammation [58,59]. Moreover, there is accumulating evidence that moderate-vigorous exercise training improves cognitive function, particularly in areas of attention-psychomotor function, executive function, and memory [60,61]. Higher cardiorespiratory fitness, as peak oxygen uptake (peak VO_2_), has been associated with higher cognitive function [62], attenuation of gray matter volume atrophy [31,63], greater white matter integrity [64], and higher MCAv and cerebrovascular function [62,65,66]. Furthermore, maintaining or increasing cardiorespiratory fitness over time has been shown to reduce the risk of dementia incidence and mortality [67].

While normal aging is associated with a decline in resting levels of CBF, habitual exercisers with higher cardiorespiratory fitness have been shown to have higher resting MCAv levels than age-matched sedentary counterparts [66]. Proposed mechanisms for this involve exercise-induced increases in the recruitment and/or new growth of capillaries (i.e., angiogenesis), as well as beneficial vascular remodelling from repetitive hemodynamic forces on the artery walls (i.e., shear-stress) that occurs with exercise [38,68]. A recent meta-analysis by Smith et al. [18], found no change in resting MCAv with exercise training interventions (2–7 months duration) compared with control. Although, seven of the eight included studies did not measure changes in cerebrovascular conductance index (i.e., MCAv/MAP), which would account for potential changes in arterial blood pressure with exercise training. In contrast, results from The Brain in Motion Study (n = 206 healthy middle-aged adults) [69] showed 6-months of moderate intensity aerobic training significantly improved resting MCAv and reduced cerebrovascular resistance in conjunction with improvements in cognitive functions. Furthermore, Tomoto et al. [70] showed 12-months of moderate-vigorous aerobic training increased total CBF (normalized for total brain mass) and reduced cerebral pulsatility in patients with mild cognitive impairment, which were associated with changes in peak VO_2_.

The effect of habitual exercise and cardiorespiratory fitness on cerebrovascular function has also been studied, by assessing cerebrovascular reactivity (i.e., changes in CBF or MCAv with changes in arterial CO_2_). Similar to resting CBF or MCAv, cerebrovascular reactivity decreases with age [13], however higher levels of habitual exercise and cardiorespiratory fitness are associated with greater cerebrovascular reactivity [71,72]. Furthermore, several studies have shown aerobic exercise training (compared with control), can improve cerebrovascular reactivity in middle-aged healthy adults [73], stroke survivors [74], and breast cancer survivors (high intensity group) [75]. Concomitantly, two of these studies demonstrated substantial improvements in peak VO_2_ (~19%) [74,75], and Ivey et al. [74] found a positive correlation (r = 0.55) between changes in peak VO_2_ and changes in cerebrovascular reactivity. In contrast, The Brain in Motion Study [69] found no improvement in cerebrovascular reactivity with 6-months of moderate intensity aerobic training. This study showed a significant, but small improvement in peak VO_2_ (~7%). Therefore, substantial improvements in cardiorespiratory fitness (as peak VO_2_) may be necessary for eliciting changes in cerebrovascular reactivity. Although, a study by Thomas et al. [76] found cerebrovascular reactivity was reduced in elderly Masters athletes compared with elderly sedentary males. The authors suggested the lower cerebrovascular reactivity may be related to vascular desensitization from chronic exposure to higher CO_2_ levels with exercise training [76]. Over a short-term period, we have found no effect of repeated hypercapnia (CO_2_) exposures on cerebrovascular reactivity in healthy adults [77].

Higher levels of cardiorespiratory fitness have also been positively associated with volumes of gray matter and white matter in older adults, specifically the attenuation of the age-related atrophy rates in the frontal, temporal, and parietal regions [63,78]. Therefore, exercise may contribute to a greater ‘reserve’ or threshold against the clinical manifestations of cognitive impairment. This concept of ‘reserve’ emerged due to individual differences in the susceptibility of cognitive impairment as well as discontinuity between cognitive outcomes and neuropathology (e.g., β-amyloid burden & neurofibrillary tangles of Tau) [31,79]. Stern [80] discusses the concept of ‘brain reserve’, whereby individuals with greater brain volumes may tolerate higher levels of neuropathology without affecting cognitive function; and ‘cognitive reserve’ whereby cognitively normal adults tolerate a higher level of neuropathology through pre-existing cognitive processes or compensatory approaches [80]. Exercise and stimulating environments may contribute to brain and cognitive reserve through the upregulation of neurotrophic growth factors such as brain-derived neurotrophic factor and insulin-like growth factor, which promote neurogenesis (i.e., the formation and maintenance of neurons) and neuroplasticity (i.e., the improvement of brain structure and function) [81,82]. While life-long exercise and higher levels of cardiorespiratory fitness are associated with less atrophy of gray matter volumes, the evidence for exercise training increasing gray matter volumes is less convincing and may be specific to the left hippocampus region [83].

## 5. Effect of Exercise Training on Brain Health in CVD

Previous reviews that have investigated the effect of cardiac rehabilitation on cognitive function [53,84] have found promising but limited evidence for exercise-based cardiac rehabilitation on cognitive function. A literature search of MEDLINE and Scopus was recently conducted by Taylor [85] identifying fourteen exercise training studies in patients with CVD and/or HF assessing brain-related outcomes (i.e., cognitive function, CBF, cerebral artery blood velocity, cerebrovascular function, and brain imaging markers). A summary of the study characteristics and brain-related outcomes for thirteen chronic exercise training studies (≥14 days) are outlined in Table 1. One study by Anazodo et al. published their results as two manuscripts including outcomes of cognitive function and brain structure [15] and CBF [14]. Twelve of thirteen exercise training studies were performed within a cardiac rehabilitation setting. Cardiac rehabilitation is internationally recognized as a class 1A recommendation for patients following a cardiovascular-related event or procedure, which promotes intensive CVD risk factor modification and exercise training [86,87].

### 5.1. Cognitive Function

Twelve of the thirteen studies assessed cognitive function [14,15,88,89,90,91,92,93,94,95,96,97,98] (Table 1). Six of nine studies found a significant improvement in global cognitive function, using the Montreal Cognitive Assessment (MoCA) [93,98], Mini-Mental State Examination (MMSE) [95] Modified MMSE [90], Functional Independence Cognitive Measure [97], and NIH Toolbox Fluid Composite Score [96]. Seven of eight studies showed improvements in at least one test related to attention-psychomotor function [88,89,90,91,92,95,96]. Only Lee et al. [94] found no improvement in attention-psychomotor function. Three of seven studies showed improvements in executive function [88,95,96], while four studies found no improvement [90,91,92,94] (Table 1). Three of five studies found an improvement in verbal memory [90,91,92] while two studies found no change [94,96]. Two of four studies showed an improvement in visuospatial working memory [90,96] and two studies showed no change [88,92] (Table 1). Finally, none of the five studies assessing language with verbal fluency or naming tests showed a significant effect of exercise training [88,89,90,91,92]. Taken together, the current evidence suggests exercise training in patients with CVD or HF improves global cognitive function and attention-psychomotor function but not language processes [85]. The effect of exercise training on executive functions and memory in patients with CVD or HF is less clear with conflicting results between studies [85].

Taylor [85] reported a major limitation within the available studies is the lack of a control group, and therefore it cannot be determined whether improvements in cognition following a cardiac-related event would occur naturally without exercise training or could be from learning effects related to the cognitive tests. Stanek et al. [90] found improvements in attention-psychomotor function and verbal memory exceeded those of practice effects but improvements in global and executive function were similar to practice effects. Tanne et al. [88] found significant improvements in measures of psychomotor and executive functions compared with controls that could not complete the exercise training intervention, but no differences for improvements in global function (MMSE) or other neurocognitive tests. Finally, Fujiyoshi and colleagues [95] found patients attending monthly cardiac rehabilitation over 6-months had significantly greater improvements in global cognition and executive function than patients attending less than monthly cardiac rehabilitation as a control group.

From the available studies it is difficult to determine whether intensity of the exercise interventions plays a role in improving cognitive functions. Six of the thirteen studies did not report details of exercise intensity. However, the majority of included studies reported significant improvements in exercise capacity for the exercise groups (Table 1). Furthermore, several studies reported significant correlations between change in exercise capacity and change in cognitive domains, including change in peak metabolic equivalents (METs) and attention-executive function [89], change in peak METs and verbal memory [90], and change in submaximal test METs and working memory [96]. The two studies that found no improvement in cognition, also found no significant improvements in peak VO_2_ [15,94].

### 5.2. Brain Structure and Cerebrovascular Function

The effect of exercise training on brain structure was only measured by one study [15]. Using magnetic resonance imaging (MRI), Anazodo et al. [15] found significant bilateral improvements in gray matter volume within the frontal lobe, middle temporal gyrus, supplementary motor area during cardiac rehabilitation, which were areas that showed significant atrophy compared with healthy controls at baseline. One study investigated the effect of exercise training on CBF using MRI arterial spin labelling [14], and three studies measured the effect on cerebral artery blood velocity using transcranial doppler ultrasound (TCD) [88,90,99]. One study also used near-infrared spectroscopy as a marker of cerebral oxygenation [96]. In the same MRI study above, Anazodo et al. [14] found significant improvements in regional gray matter CBF bilaterally within the Anterior Cingulate by 30%, but no significant change in global CBF. Of note, the Anterior Cingulate lies within the medial aspect of the frontal lobe and is known to have an important role in executive functions relating to emotional control, focused problem-solving, error recognition, and adaptive responses to changing conditions [100]. Using TCD, none of the three studies [88,90,99] found a significant change in resting MCAv, and Smith et al. [99] found a decrease in resting posterior cerebral artery velocity (PCAv). However, Stanek et al. [90] found a significant improvement in resting anterior cerebral artery velocity (ACAv). Notably, the ACA branches into the pericallosal artery to supply CBF to the Anterior Cingulate [101], where Anazodo et al. [14] found a significant increase in CBF. Stanek et al. [90] also found that higher ACAv and MCAv at baseline was associated with greater improvements in visuospatial working memory.

In terms of cerebrovascular regulation, Tanne et al. [88] measured MCAv cerebrovascular reactivity with TCD using a breath-hold index and found no significant improvements in either the exercise group or control. It is intriguing that this study found improvements in cardiovascular hemodynamics (cardiac index and systemic vascular resistance) but this did not translate into improvements in MCAv or cerebrovascular reactivity in patients with HF [88]. This study also demonstrated significant improvements in exercise capacity with 6-min walk test and treadmill METs, however changes in cardiorespiratory fitness as peak VO_2_ were not directly measured. Smith et al. [99] measured cerebrovascular regulation during submaximal and maximal exercise before and after 12-weeks of exercise-based cardiac rehabilitation in patients with left ventricular assist devices (LVADs). Despite a reduction in resting PCAv following exercise training, patients with LVAD achieved greater increases in PCAv during submaximal and maximal exercise, compared with before training. Similarly, following exercise training patients had higher flow through the internal carotid artery during submaximal exercise, although this did not lead to improved regulation of MCAv. Overall, there is very limited evidence available assessing the effect of exercise training on brain structure and cerebrovascular function in patients with CVD.

## 6. Is There Rationale for Higher Intensity Exercise?

The magnitude of change in cardiorespiratory fitness [89,90,96] and vascular function [102] during cardiac rehabilitation has been shown to significantly correlate with changes in cognitive function. High intensity interval training (HIIT) is a potent stimulus for improving cardiorespiratory fitness (as peak VO_2_) and vascular function compared with moderate intensity continuous training (MICT) [103,104,105]. Importantly, the superior effects of HIIT on peak VO_2_ and vascular function compared with MICT also occurs in patients with CVD [106,107,108,109], and HIIT has been shown to be safe and feasible in patients attending cardiac rehabilitation [110,111,112]. Furthermore, in patients with HF, HIIT has shown greater improvements in cardiac function compared with MICT [106,113]. Therefore, we propose that greater improvements in cardiorespiratory fitness, vascular function, and cardiac perfusion with HIIT, has potential to translate into greater adaptations for cerebrovascular health and cognitive function (see Figure 1).

HIIT involves alternating bouts of high intensity exercise interspersed with bouts of active or passive recovery. Various protocols of HIIT have been studied in clinical and healthy populations, ranging from short-duration HIIT protocols (work intervals < 1-min), medium-duration HIIT protocols (work intervals 1–3 min), and long-duration HIIT protocols (work intervals ≥ 3 min) [114,115]. Furthermore, HIIT protocols accumulating ≥ 15-min of high intensity exercise in total have been defined as high-volume HIIT [105]. HIIT should not be confused with sprint interval training, which instead involves brief maximal or supramaximal efforts (≥100% peak oxygen uptake or work capacity) typically with shorter work intervals than HIIT and a greater proportion of recovery time [103]. A well-known example of a high-volume HIIT protocol is the 4x4 Norwegian protocol involving four bouts of 4-min high intensity intervals (at ~85–95% of peak heart rate; hard to very hard effort) interspersed with 3-min lower intensity recovery intervals (at ~60–70% of peak heart rate; fairly light effort) [109,116,117,118]. High-volume HIIT protocols have demonstrated superior improvements on health outcomes compared with MICT (particularly for peak VO_2_) while low-volume HIIT protocols typically show comparable improvements with MICT [105,115]. A HIIT protocol can be applied to various forms of exercise based on an individual’s preference and capabilities, such as walking hills, jogging, cycling, rowing ergometer, elliptical, swimming, or aerobics/dance [119].

A proposed mechanism for the superior effect of HIIT on peripheral vascular function, when compared with MICT, is that higher intensity exercise provokes greater blood flow and shear stress stimulus that allows for greater vascular adaptation and upregulation of vasodilatory prostaglandins [13,38] and nitric oxide [38,120]. While this evidence has emerged mainly from studies of the forearm artery [104], it is reasonable to infer that greater shear stress and therefore vascular adaptation, would also occur in the cerebral vasculature. Indeed, Ogoh et al. [121] showed significantly higher internal carotid artery velocity (i.e., shear stress) for HIIT compared with MICT during acute exercise and recovery. Moreover, Klein et al. [122] found significantly greater total MCAv (i.e., shear stress) during an acute bout of HIIT compared with MICT for younger and older adults. Historically our understanding of CBF changes with cycle exercise has involved an inverted U-shape pattern, whereby MCAv and internal carotid artery flow increases with exercise intensity to ~60% of maximal VO_2_ and then decreases toward baseline levels with higher intensities due to hyperventilation-induced hypocapnia (reduced CO_2_) and cerebral vasoconstriction [123]. Therefore, exercise at higher intensities seemed counterintuitive to increasing CBF and shear stress within the cerebral circulation [124]. However, combining intervals of high intensity exercise with short intervals of active recovery (i.e., HIIT) appears to lead to a greater accumulation of CBF and shear stress than MICT [121,122]. Furthermore, studies examining MCAv during rowing exercise [125] and running [126] have shown that MCAv may not exhibit the same inverted U-shape pattern as cycling exercise. With running, Furlong et al. [126] showed a continual increase in MCAv across the full range of intensities up to 95% VO_2_max, compared with the usual inverted U-shape pattern for cycling (peaking at 65% VO_2_max). Faull et al. [125] found similar patterns of MCAv response with increasing intensities of rowing exercise, although this was not compared with cycling exercise. Taken together, these studies suggest acute bouts of higher intensity exercise provide a greater dose of shear stress, that may promote greater cerebrovascular adaptations with exercise training.

There are no published studies examining the effect of HIIT on cerebrovascular outcomes in patients with CVD. In non-cardiac populations, Whittaker et al. [127] conducted a systematic review on the acute and chronic effects of HIIT on cerebrovascular outcomes. Only three studies assessed the chronic effect of HIIT on cerebrovascular outcomes, including cerebral oxygenation using NIRS in healthy older adults [128], dynamic cerebral autoregulation in endurance training males [129], and cerebrovascular reactivity in breast cancer survivors [75]. Overall, none of the studies found a significant interaction effect between time and exercise intensity on any cerebrovascular outcomes, however it should be highlighted that all studies were small and not adequately powered to detect significant differences between groups. The pilot study by Northey et al. [75] in breast cancer survivors showed that 12-weeks of HIIT resulted in moderate-large positive effects on resting MCAv (effect size (ES) = 0.86) and MCAv cerebrovascular reactivity (ES = 0.72) compared with control, and a moderate effect on MCAv cerebrovascular reactivity (ES = 0.54) compared with MICT. Drapeau et al. [129] found a subtle reduction in dynamic cerebral autoregulation after 6-weeks of HIIT in endurance training athletes, although there was no comparison to a moderate intensity group or control group.

To date, only one study by Lee et al. [94] has compared the effect of HIIT and MICT on cognitive function during cardiac rehabilitation and found no significant improvement in either group for any cognitive domains (Table 1). Although, recruitment and retention challenges resulted in a small sample size (n = 7 per group) which was underpowered to detect differences between groups. This study also found no significant group differences for peak VO_2_ improvement between HIIT and MICT despite using a high-volume 4 × 4-min HIIT protocol (i.e., greater dose than MICT); and the improvement in peak VO_2_ with HIIT (1.3 mL/kg/min) was 3-fold lower in comparison to the peak VO_2_ improvements with HIIT in other studies involving patients with coronary artery disease (mean = 4.6 mL/kg/min) [106]. A reason for this may be that most of the exercise training was performed in a home-based setting with one supervised exercise session per week. Several small studies have compared HIIT and MICT on cognitive function in non-cardiac populations. A study in older adults [130] found improvement in working memory with 12-weeks of HIIT but not MICT. In younger adults [131], 6-weeks of HIIT but not MICT showed improvement in executive function response time. The pilot study by Northey et al. [75] comparing HIIT and MICT with no-exercise control in breast cancer survivors, found moderate-large positive effects for HIIT in comparison to MICT for executive function (ES = 0.55) and working memory (ES = 1.41) and in comparison to no-exercise control for episodic working (ES = 0.76), executive function (ES = 0.75), and working memory (ES = 0.81). Of note, the improvement in peak VO_2_ significantly increased with HIIT (19%) but not MICT (6%) or control (3%), which supports the notion that the stimulus of the intervention on cardiorespiratory fitness is likely important for changes in cognitive functions and cerebrovascular outcomes. More studies are needed comparing exercise of different intensities on measures of cognitive function, cerebrovascular outcomes, and brain structure.

## 7. Conclusions

Patients with CVD and HF are at increased risk of cognitive decline. Exercise training, predominantly during cardiac rehabilitation, consistently shows improvements in global and attention-psychomotor cognitive functions, but the effect on executive function and memory is less clear. Although, an important caveat is the lack of control groups and therefore cannot be definitive that improvements did not occur from practice effects or naturally with time following a cardiac event. However, the degree of cardiorespiratory fitness improvement with exercise training appears to be an important mediator (or marker of adequate training stimulus) for changes in cognitive function and cerebrovascular outcomes in healthy and clinical populations. Since HIIT is a potent stimulus for improving cardiorespiratory fitness, this supports the hypothesis that HIIT may also promote superior improvements in cognitive function and cerebrovascular outcomes. Furthermore, the greater accumulation of CBF and shear stress during HIIT compared to MICT, may allow for greater vascular adaptations and improvements in cerebrovascular outcomes and cognitive function. Currently, there is a very limited amount of research investigating the influence of exercise training intensity on cognitive function and cerebrovascular outcomes, and there are no studies in CVD or HF populations. Well-designed exercise training studies are warranted to investigate the effect of HIIT on cognitive function and cerebrovascular outcomes in patients with CVD and HF; and determine the optimal exercise prescription for improving brain health in this high-risk population.

## Figures and Tables

**Figure 1 ijerph-19-16926-f001:**
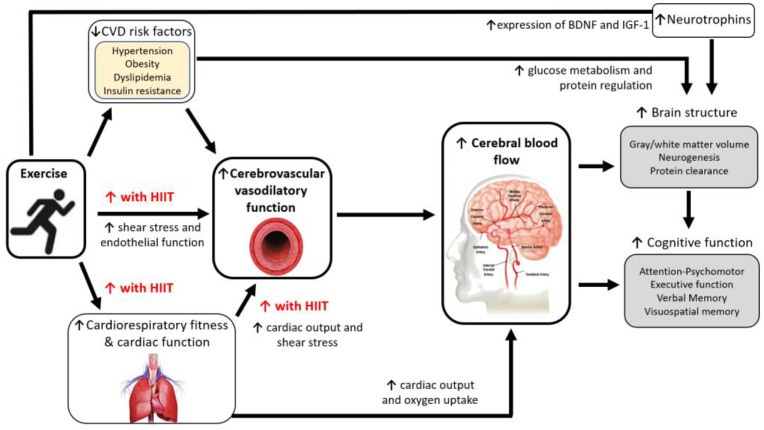
Potential mechanisms for how high intensity exercise can improve brain health. Abbreviations: BDNF: brain-derived neurotrophic factor; CVD: Cardiovascular disease; HIIT: high intensity interval training; IGF-1: insulin-like growth factor 1; ↑: increase, ↓: decrease.

**Table 1 ijerph-19-16926-t001:** Chronic Exercise Training Studies in Patients with CVD Assessing Change in Brain-related Outcomes.

Study, Year, Design, and Sample Size	Patient Characteristics	Exercise Intervention	Change in Exercise Capacity	Summary of Brain-Related Outcome Results
Tanne et al. [88] (2005) Controlled Trial; Ex = 20, Control = 5	HF—NYHA III 63 ± 13 years; 20% F BMI: 26.7 EF: 26 ± 5% 6 MWT: 308 ± 87 m	18-week CR; 2 d/w Aerobic; 50 min 60–70% HRmax	↑ 6 MWT distance (MD = 115 m) ↑ Modified Bruce Test (MD = 4.2 min)	**Cognition:** (+) Improved Attention-Psychomotor (TMT-A) and Executive functions (TMT-B, Stroop-A) for Ex group only. (−) NSC in Global function (MMSE), Visuospatial memory (Rey-Osterrieth Complex Figure), Language (Verbal fluency), and Stroop B + C or Continuous performance test. **MCAv (TCD):** (−) NSC in MCAv & CVR breath-hold index **Other Hemodynamics**: (+) Improved CV hemodynamics (↑ cardiac index & ↓ systemic vascular resistance).
Gunstad et al. [89] (2005) Case-Series; Ex = 18, no Control	CAD/CABG/HF 68 ± 8 years; 28% F METs: 5.2 ± 2.0	12-week CR; 3 d/w Aerobic; 10–45 min Intensity NR	↑ Peak METs on clinical stress test (MD = 2.2)	**Cognition**: (+) Improved Attention-Psychomotor function (TMT-A & DSC) (−) NSC in GP & Language (Animal fluency).
Stanek et al. [90] (2011) Case-Series; Ex = 51, no Control	CAD/CABG/HF 68 ± 9 years; 35% F BMI: 31.3 METs: 7.2 ± 2.7	12-week CR; 3 d/w Aerobic; 60 min Intensity NR	↑ Peak METs (MD = 2.8)	**Cognition:** (+) Improved Attention-Psychomotor function (LNS, GP) & Verbal Memory (HVLT) > practice effects. (+) Improved Global function (Modified MMSE) and Visuospatial Memory (BVMT) = practice effects. (−) NSC Executive function (TMT, FAB) or Language (Animal fluency, Boston naming). **MCAv and ACAv (TCD)**: (+) ↑ACAv. (−) NSC in MCAv or pulsatility index.
Anazodo et al. [15] (2013); Case-Series; Ex = 24, no Control Anazodo et al. [14] (2016); Ex = 17	CAD 59 ± 6 years; 25% F BMI: 29.8 ± 4.7 EF: 64 ± 8% PeakVO_2_: 26 ± 2	6-month CR; 3 d/w Aerobic + RT 20–30 min RPE 11–14 40–70%HRR	NSC PeakVO_2_ (MD = 5% increase, 1.3 mL/kg/min)	**Cognition**: (−) NSC MoCA score. **Brain structure:** (+) ↑ Gray matter volume bilaterally in frontal lobe, middle temporal gyrus, supplementary motor area. **CBF (ASL)**: (+) ↑ Regional gray matter CBF in bilateral Anterior Cingulate. (−) NSC in Global Gray Matter CBF.
Alosco et al. [91] (2014) Case-Series; Ex = 52, no Control	HF—NYHA II/III 67 ± 8 years; 25% F EF: 39 ± 12% 2-min step test: 72 ± 20	12-week CR; 3 d/w Aerobic; 40 min Intensity NR	↑ 2-min step test over 12-wk (MD = 4) NSC 2-min step test over 12-months (MD = 5)	**Cognition**: (+) Improved Attention-Psychomotor function (DSC) at 12-wks & 12-months. (+) Improved Verbal Memory (CVLT-II) at 12-months but NSC at 12-wks. (−) NSC in Global function (MMSE), Executive function (TMT-B), Language (Animal Fluency, Boston Naming), or TMT-A.
Santiago et al. [92] (2018) Case-Series; Ex = 50, no Control	ACS 67 ± 7 years; 16% F BMI: 28.6 ± 4.2 PeakVO_2_: 19 ± 5	48-week CR; 1 d/w + 4 d/w at home Aerobic + RT Walk/jog; Intensity NR	↑ PeakVO_2_ (MD = 28%, 5.3 mL/kg/min)	**Cognition:** (+) Improved Attention-Psychomotor function (TMT-A, DSC) & Verbal memory (CVLT-II). (−) NSC in Executive function (TMT-B), Visuospatial memory (BVMT-R), & Language (Animal fluency, FAS Verbal fluency).
Salzwedel et al. [93] (2019) Cohort Study; Ex = 401, no control	ACS/CABG 55 ± 6 years; 20% F BMI 28.7 ± 5.1	3-week CR Exercise details NR	↑ 6 MWT distance (MD = 83 m)	**Cognition:** (+) Improved MoCA score.
Lee et al. [94] (2019) RCT; Ex1 = 7, Ex2 = 7, no control	CAD/PCI/CABG 68 ± 9 years; 100% F BMI 28 ± 6 EF: >35% PeakVO_2:_ 19 ± 4	24-week CR; 5 d/w Aerobic walk/jog Ex1: 30–40 min 60–80% PeakVO_2_ Ex2: 4 × 4 min 90–95% PeakHR RPE > 17 3 d/w + MICE 2 d/w	↑ PeakVO_2_ for Ex2 (MD = 7%; 1.3 mL/kg) NSC PeakVO_2_ for Ex1 (MD = 2% 0.4 mL/kg). NS difference between groups.	**Cognition**: (−) NSC in Attention-Psychomotor function (DSC), Executive function (TMT-B, Digit span test), and Verbal Memory (CVLT-II) for moderate intensity (Ex1) or aerobic interval training (Ex2) groups.
Fujiyoshi et al. [95] (2020) Controlled Trial; Ex1 = 27, Ex2 = 39	CVD 77 ± 5 years; 44% F BMI: 24 ± 3 y 6 MWT: 465 ± 98 m	6-month CR Aerobic + RT BORG 10–13 Ex1: 1/month Ex2: <1/month	↑ 6 MWT distance for Ex1 (MD = 40 m) NSC 6 MWT distance for Ex2 (MD = −14 m)	**Cognition**: (+) Improvement in Global (MMSE) and Executive functions (FAB) was significantly greater monthly CR (Ex1) than <monthly CR (Ex2), specifically improved were temporal orientation, attention, calculation, No/No-Go task). **Vascular function: **(+) Improvement in vascular function (as reactive hyperemia peripheral arterial tonometry) was significantly greater with monthly CR (Ex1) than the <monthly CR (Ex2).
Moriarty et al. [96] (2020) Case-Series; Ex = 20, no Control	CVD 65 ± 12 years; 25% F BMI: 29 ± 6 y METs: 5.5 ± 2.5	6-week CR 3 d/w, 30–60 min 50–80%HRR RPE 3–5/10	↑ METs from submaximal treadmill test (MD = 1.4)	**Cognition**: (+) Improved Global function (NIH Toolbox Fluid Composite score), and specifically Attention, Processing Speed, Executive Function, Visuospatial Working Memory. (−) NSC in Episodic (Verbal) Memory. **Cerebral oxygenation (NIRS):** (+) Improved oxygenation of right and left pre-frontal cortex during cognitive testing
Sumida et al. [97] (2020); Cohort Study Ex = 111, no Control	ACS/CABG/HF Age: 77 years BMI: 22	2–6 week inpatient CR; 2–3 d/w; Intensity NR	↑ FIM-Physical score	**Cognition**: (+) Improved Functional Independence Measure (FIM-Cognitive).
Redwine et al. [98] (2020); RCT; Ex1 = 24, Ex2 = 22, Control = 23	Symptomatic HF 65 ± 10 years; 11% F LVEF: 46 ± 14%	16-week; 2 d/w; 60 min; RPE 11–13 Ex1: Tai Chi; Ex2:RT	None reported	**Cognition**: (+) Improved MoCA score for Ex groups (Tai Chi & RT) compared with Control.
Smith et al. [99] (2020) Case-Series; Ex = 12, no Control	LVAD 54 ± 12 years; 42% F PeakVO_2:_ 12 ± 3	12-week CR 3 d/w, 60 min Aerobic + RT 59–90% VO_2_R; RPE 11–15; 50–60%1 RM	↑ PeakVO_2_ (MD = 25%; 2.9 mL/kg)	**Cognition**: Not assessed. **MCAv and PCAv (TCD):** (+) Improved PCAv regulation during maximal exercise following the training period. Improved ICA flow regulation during submaximal exercise following training period. NSC in MCAv regulation. (−) Decrease in resting PCAv. NSC in resting MCAv.

**Abbreviations:** ACAv: Anterior cerebral artery velocity; ACS: Acute Coronary Syndrome; ASL: Arterial-spin Labelling; BVMT: Brief Visual Memory Test; CABG: Coronary artery bypass graft surgery; CAD: Coronary Artery Disease; CR: Cardiac rehabilitation; CVD: Cardiovascular disease; CVLT: California Verbal Learning Test; DSC: Digit Symbol Coding Test; Ex: Exercise group; FAB: Frontal Assessment Board; GP: Grooved Pegboard; HF: Heart Failure; HVLT: Hopkins Verbal Learning Test; HR: Heart rate; HRR: Heart rate reserve; ICA: Internal carotid artery; LNS: Letter-number Sequencing; LVEF: Left ventricular ejection fraction; MCAv: Middle cerebral artery velocity; MD: Mean Difference; METs: Metabolic equivalents; MoCA: Montreal Cognitive Assessment; NIRS: Near-infrared Spectroscopy; NR: Not reported; NSC: non-significant change; PeakVO_2_: Peak Oxygen Uptake; PCAv: Posterior cerebral artery velocity; RCT: Randomized Controlled Trial; RM: Repetition maximum; RT: Resistance training; SVR: systemic vascular resistance; TCD: Transcranial doppler ultrasound; TMT: Trail-making Test; LVAD: left ventricular assist device; VO_2_R: oxygen uptake reserve; ↑: increase; ↓: decrease.

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
