# Peer review of "The Utility of High Intensity Interval Training to Improve Cognitive Aging in Heart Disease Patients"

_ijerph, 2022, doi:10.3390/ijerph192416926_

Round 1

Reviewer 1 Report

This review compiles information related with the possible beneficial effects of high intensity training on cognitive complications in heart disease patients. Authors present relevant information supporting th importance of high intensity training and how this may improve cardio respiratory fitness and cardiac and cerebrovascular function increasing cerebral blood flow leading to an improvement of cognitive function.

Minor comments:
- Correct the format of cited References 58, 86 in first sentence of section: Effect of Exercise Training on Brain Health in CVD. Page 5.

- First sentence on section: Cognitive function mentions twelve studies but in the table 14 studies are cited.

- I suggest that the Information describing the table and the information shown in the table be written more clearly.

- Authors should check the Figure legend.

Reviewer 2 Report

The manuscript by Jenne L. Taylor and colleagues is an elegant, logically structured review article that discusses the relationship between regular physical activity, cardiovascular disease and cognitive function in health and disease, with a list of relevant scientific evidence. The paper emphasizes that though HIIT interval training may be beneficial for the maintenance of cerebral blood flow and brain health via improving cardiorespiratory fitness and cerebrovascular function, there are only few studies investigating the link between exercise intensity and cognitive function and brain health.

I only have a few minor suggestions, otherwise I recommend publishing the article.

·       In Chapter 2,  I recommend to devote a few sentences to the fact that CBF regulating processes also participate in the maintenance of blood brain barrier integrity, the loss of which is a relevant mechanism in the development of dementia.

·       On page 3, paragraph 2: „The relationship between cardiac output and CBF appears to be linear” should be rephrased, as most of the referred studies investigated the relationship between cardiac output and cerebral artery flow velocity, rather than CBF itself. The relationship between CO and CBF is also affected by changes in cerebral vascular resistance (espesially in normal blood pressure ranges in healthy individuals).

·       On page 3, paragraph 3: „Arterial stiffness and reflected blood flow back to the heart…” is not completely correct. It is not exactly blood flow what returns, but rather the reflection of the pulse wave, which is a pressure wave.

·       Figure 1. has no caption.

Reviewer 3 Report

Dear Authors

Thank you for the opportunity to review your manuscript with the topic: “The Utility of High Intensity Interval Training to Improve Cognitive Aging in Heart Disease Patients

This is a highly topical subject of great relevance that fits in the scope of the International Journal of Environmental Research and Public Health.

The submitted review is well written and directs very well to the topic. The discussion is thorough and also states how difficult it is to draw definite conclusions from the type of training performed to the improvement in global brain function as the ultimate goal. Therefore, I consider it a strength of this review to elaborate on "intermediate goals" such as improvement in CBF or improvement in grey matter volume etc, even if this makes the manuscript longer. That these improvements do not necessarily improve cognitive aging as suggested by the title is sufficiently discussed and understandable for the reader.

Therefore I recommend to publish this review in IJERPH.

Nevertheless, I have some minor points that should to be addressed

Unfortunately the numbering of the lines starts at page 10

P5 3rd§ L2: … cognitive function 52,86 have found… : Quotation in [ ] and not superscript

P10 L38: …oxygena-tion[88] space is missing

P11 L57: …HF [86].   This study..  double space

P11 “6. Is there a rationale for higher intensity exercise”

As HIIT is the main focus of this manuscript I recommend to explain how this special training is performed in detail. What kind of activity is performed for how long with what intensity how long are the breaks etc.?

To get the reader a vividly imagine of HIIT

P11 L81: (see Figure): 1 is missing

P13 L165: “than” should be replaced by “to”

References:

Standard Journal Abbreviations (ISO4) should be used consistently, e.g.:

J Am Coll Cardiol

J Appl Physiol

Front Aging Neurosci

J Neurosci Methods

and so on
